# Divergent neural circuits for proprioceptive and exteroceptive sensing of the *Drosophila* leg

Su-Yee J. Lee[1], Chris J. Dallmann[1,2], Andrew Cook[1,3], John C. Tuthill [1] ✉ & Sweta Agrawal [1,3] ✉

Somatosensory neurons provide the nervous system with information about mechanical forces originating inside and outside the body. Here, we use connectomics from electron microscopy to reconstruct and analyze neural circuits downstream of the largest somatosensory organ in the *Drosophila* leg, the femoral chordotonal organ (FeCO). The FeCO has been proposed to support both proprioceptive sensing of the fly's femur-tibia joint and exteroceptive sensing of substrate vibrations, but it was unknown which sensory neurons and central circuits contribute to each of these functions. We found that different subtypes of FeCO sensory neurons feed into distinct proprioceptive and exteroceptive pathways. Position- and movement-encoding FeCO neurons connect to local leg motor control circuits in the ventral nerve cord (VNC), indicating a proprioceptive function. In contrast, signals from the vibration-encoding FeCO neurons are integrated across legs and transmitted to mechanosensory regions in the brain, indicating an exteroceptive function. Overall, our analyses reveal the structure of specialized circuits for processing proprioceptive and exteroceptive signals from the fly leg. These findings are consistent with a growing body of work in invertebrate and vertebrate species demonstrating the existence of specialized limb mechanosensory pathways for sensing external vibrations.

To coordinate complex behaviors, circuits in the central nervous system (CNS) require continuous information about the body and the environment. Somatosensory neurons are an important source of feedback that provide the nervous system with information about mechanical forces acting on an animal's body[1,2]. Neurons of the somatosensory system are typically described as either exteroceptive, detecting mechanical forces generated in the external world, or proprioceptive, detecting the position or movement of body parts. However, because they are embedded within the body, many somatosensory neurons can detect both externally- and self-generated forces, making it difficult to determine whether specific somatosensory neurons are exteroceptive, proprioceptive, or both. Recording from

primary somatosensory neurons in behaving animals can resolve the types of mechanical stimuli they encode[3], but such experiments are technically difficult and often not feasible. An alternative approach is to map the connectivity of sensory neurons with downstream circuits, which can provide clues about putative function. For example, proprioceptor axons often synapse directly onto motor neurons to support rapid reflexes[4] (Fig. 1A). In contrast, exteroceptive signals are often integrated with other sensory cues and modified by internal states to more flexibly control action selection[5,6].

Mapping the flow of sensory signals into the nervous system has recently become feasible in small organisms thanks to advances in serial-section electron microscopy (EM) and computational image

[1]Department of Neurobiology and Biophysics, University of Washington, Seattle, WA, USA. [2]Neurobiology and Genetics, Julius-Maximilians-University of Würzburg, Würzburg, Germany. [3]School of Neuroscience, Virginia Tech, Blacksburg, VA, USA. ✉e-mail: tuthill@uw.edu; sweta@vt.edu

**Fig. 1 | Connectomic reconstruction of axonal projections from somatosensory neurons in the femoral chordotonal organ (FeCO) of a female *Drosophila*.**
**A** Schematic of local and ascending VNC circuits for leg somatosensation and motor control. **B** Left: Confocal image of a *Drosophila* front leg showing the location of FeCO cell bodies and dendrites. Green: GFP; gray: cuticle autofluorescence. Right: Schematic showing the fly brain and ventral nerve cord (VNC).
**C–I** Anatomical and functional subtypes of somatosensory neurons in the *Drosophila* FeCO. **C** Calcium signals from FeCO axons of each subtype (GCaMP, black traces) in response to a controlled movement of the femur-tibia joint (gray traces). Reprinted from Neuron, vol. 111, Mamiya, A. et al., Biomechanical origins of proprioceptor feature selectivity and topographic maps in the *Drosophila* leg, 3230-3243.e14, Copyright (2023), with permission from Elsevier. **D** Calcium signals from

FeCO axons of each subtype (GCaMP, black traces) in response to an 800 Hz vibration of the femur-tibia joint (gray traces). **E** Confocal images of the axons of each FeCO subtype in the fly ventral nerve cord (VNC). Green: GFP; magenta: neuropil stain (nc82). Adapted from Agrawal et al.[25]. A: anterior; L: lateral.
**F** Reconstructed FeCO axons from each subtype in the front left leg neuromere of the FANC connectome (from left to right, *N* = 8, 13, 9, 13, 37 neurons). **G** Single reconstructed axons from each FeCO subtype in the front left leg neuromere of the FANC connectome. **H** Locations of all input synapses received by each FeCO subtype (i.e., postsynaptic sites). *n* indicates the number of synapses. **I** Locations of all output synapses made by each FeCO subtype (i.e., presynaptic sites). *n* indicates the number of synapses. Source data are provided as a Source Data file.

segmentation, which enable the reconstruction of synaptic wiring diagrams, or connectomes. Some of the most comprehensive wiring diagrams reconstructed to date include the brain and ventral nerve cord (VNC) of the adult fruit fly, *Drosophila melanogaster*[7–11]. Analysis of fly brain connectome datasets has already produced important insight into the organization and function of sensory organs on the head. For example, mapping the projections of mechanosensory neurons from the fly's antennae into the brain revealed the organization of circuits that support song detection, antennal grooming, and escape[12,13]. Volumetric EM datasets of the fly VNC[8,9], which is analogous to the vertebrate spinal cord, now make it possible to reconstruct and analyze the function of somatosensory signals from other parts of the fly's body, including the legs and wings.

Here, we take advantage of two separate connectome datasets that together span the CNS of a fruit fly, the Female Adult Nerve Cord (FANC)[8,14] and the Full Adult Fly Brain (FAFB), which was reconstructed as part of FlyWire[7,15]. We use connectomic analyses of brain and VNC circuits to investigate the largest somatosensory organ in the *Drosophila* leg, the femoral chordotonal organ (FeCO) (Fig. 1B). The cell bodies and dendrites of the FeCO are located in the femur of each leg and their axons project into the VNC (Fig. 1B)[16–19]. The *Drosophila* FeCO is comprised of ~150 excitatory (cholinergic) sensory neurons that can be separated into five functionally and anatomically distinct subtypes: (1) extension- and (2) flexion-encoding claw neurons encode tibia position, (3) extension- and (4) flexion-encoding hook neurons encode tibia movement, and (5) club neurons encode bidirectional tibia movement and low-amplitude (<1 μm), high-frequency tibia vibration (Fig. 1C, D)[18,19]. Claw, hook, and club neurons are named after the shape of their axons in the VNC (Fig. 1E).

The FeCO is typically described as a proprioceptive organ that monitors the movement and position of the femur-tibia joint[18–20]. However, behavioral evidence suggests that it may also detect externally-generated substrate vibrations, perhaps to aid in social communication, predator detection, and courtship[21–24]. It is currently unknown to what degree the five subtypes of FeCO sensory neurons are specialized to support specific proprioceptive or exteroceptive functions. The club neurons are the only FeCO subtype that respond to tibia vibration (Fig. 1D), suggesting that they could support exteroceptive vibration sensing[18,19]. However, club neurons also respond to larger tibia movements like those that occur during walking (Fig. 1C), suggesting that they could also be proprioceptive. Intracellular recordings from second-order neurons have identified distinct pathways for proprioceptive and vibration sensing, but in some cases also revealed complex pooling of signals from multiple FeCO subtypes[25,26].

Here, we use the FANC[14] and FlyWire[7,15] connectome datasets to reconstruct and analyze neural circuits downstream of the FeCO of the fly's front left (T1L) leg. We find that position- and movement-encoding claw and hook neurons connect to local circuits within the VNC for leg motor control, confirming their proprioceptive function. In contrast, vibration-encoding club neurons connect to intersegmental and ascending circuits that integrate mechanosensory information from the legs, wings, and neck, and relay it to the brain. By identifying these ascending projections within the FlyWire connectome, we find neurons within the brain that integrate leg vibration information with mechanosensory information from the antennae, indicating an exteroceptive function for club neurons. We also identify sparse pathways that mediate interactions between proprioceptive and exteroceptive circuits, revealing how vibration signals may directly influence motor output. Overall, our analyses suggest that the FeCO supports both proprioceptive and exteroceptive functions, which are achieved via specialized somatosensory neurons connected to specialized downstream circuits.

## Results

### Reconstruction and identification of FeCO axons in the FANC connectome

Using software for collaborative proofreading and visualization of the FANC EM dataset (see Methods), we reconstructed the anatomy and synaptic connectivity of FeCO axons from the front left leg. We focused our reconstruction efforts on these FeCO axons because they project to the front left neuromere of the VNC (also referred to as left T1 or T1L), the region of the *Drosophila* VNC with the most complete information about leg sensorimotor circuits. All of the motor neurons controlling the muscles of the front left leg and their presynaptic partners have been previously identified and reconstructed in FANC[14,27], and prior neurophysiological recordings of FeCO axons and their downstream targets were made from the front legs[18,19,25,26,28,29]. Unfortunately, leg sensory axons are among the most difficult neurons to reconstruct in all available VNC connectome datasets, likely due to rapid degeneration that begins when the legs are dissected away from the VNC during sample preparation. Leg sensory neurons have consistently darker cytoplasm and more fragmented cell membranes, leading to poor automatic neuron segmentations and synapse predictions. As a result, we reconstructed roughly half[18,19] of the FeCO axons from the front left leg (80 total axons, Fig. 1F–I). For comparison, the other publicly available VNC connectome dataset, MANC (v.1.2.1), had only 22 T1L FeCO axons reconstructed, and many of these were incomplete and missing branches. We found that the number of novel postsynaptic partners decreased as we added more axons to the dataset (Fig. S1A), suggesting that our reconstruction, while incomplete, provides a representative sample of the postsynaptic circuitry.

The FeCO consists of five functional subtypes (Fig. 1C, D)[19]. We sorted the reconstructed FeCO axons into these functional subtypes based on axon morphology and comparison with light microscopy images (Fig. 1E–G; see "Methods"). Based on measurements of the number of FeCO cell bodies in the leg[19], we estimate that we reconstructed ~50% of the T1L axons of each subtype (Supplementary Table 2). EM reconstructions of axons from each subtype qualitatively matched previous light-level images[18], including 5 club axons from the T1L leg that send an ascending projection to the brain. As expected for sensory neurons, all FeCO axons have more presynaptic sites (i.e., output synapses) than postsynaptic sites (i.e., input synapses) (Fig. 1H, I, Fig. S1C–E). Generally, the locations of pre- and postsynaptic sites are intermingled; FeCO axons do not have distinct pre- and postsynaptic zones. Finally, we do not find strong evidence for functional specialization of the different sub-branches of hook or claw neurons. Most postsynaptic neurons receive input from multiple branches (Fig. S2).

### Claw and hook (but not club) axons provide feedback to local leg motor circuits

To investigate pathways downstream of the different FeCO subtypes, we reconstructed the anatomy and synaptic connectivity of all postsynaptic partners that receive at least 4 synapses from a FeCO axon, a threshold found by previous studies to mitigate the inclusion of false positives due to errors in synapse prediction[7,30] and bias analyses towards stereotyped connections that consistently appear across multiple datasets[15]. We classified all postsynaptic VNC neurons into six morphological classes: (1) descending and (2) ascending neurons that connect the brain and VNC, (3) intersegmental neurons, which span multiple VNC neuromeres, (4) local neurons located entirely in the T1L neuromere, (5) motor neurons, and (6) sensory neurons (Fig. 2A). We interpret the connectivity between FeCO axons and local interneurons or leg motor neurons as suggesting a role in local, rapid feedback control of leg motor output. In contrast, we interpret connectivity with ascending neurons as suggesting a role in mediating sensation and behavior on longer timescales, such as sensory perception and action selection.

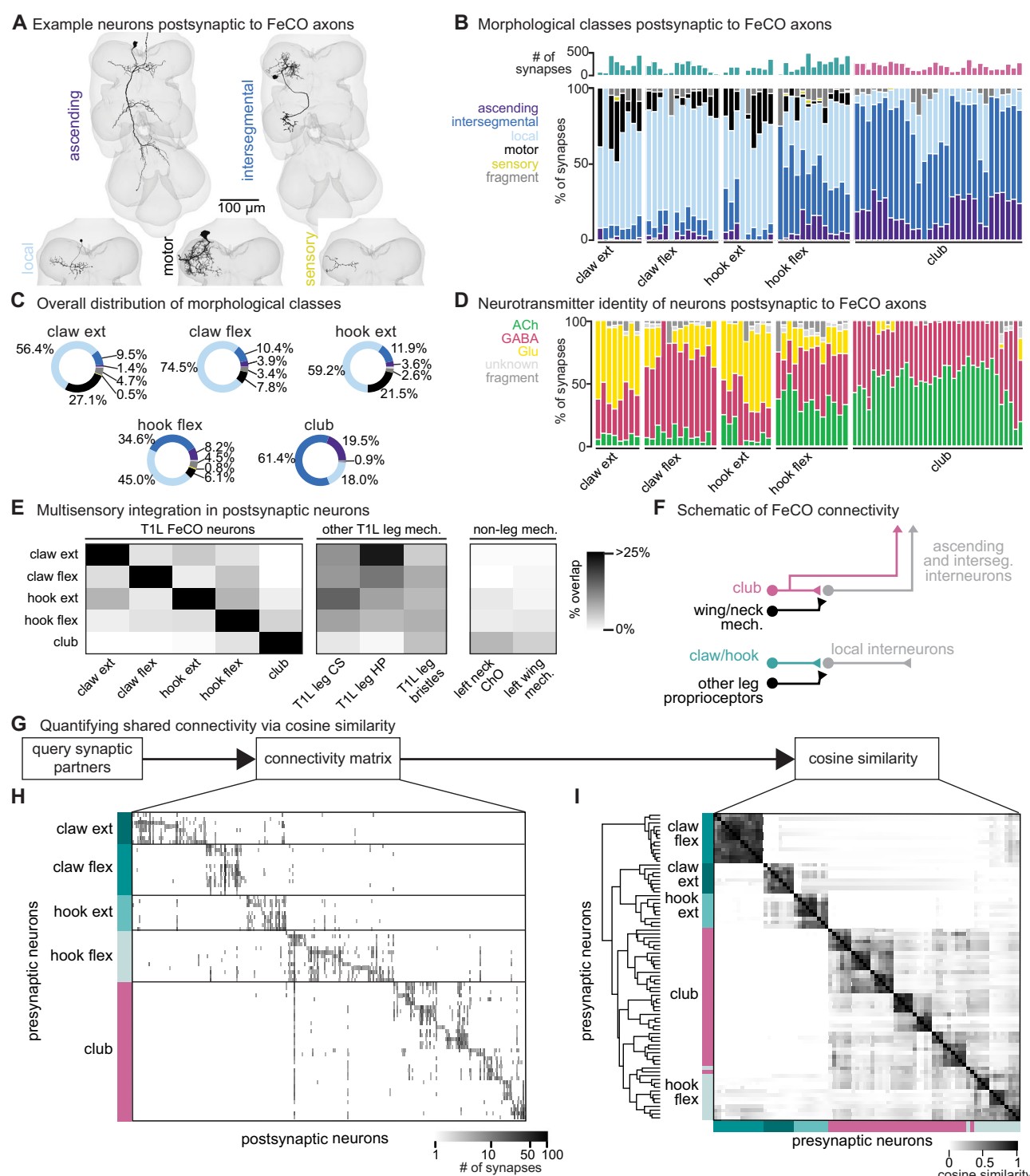

**A** Example neurons postsynaptic to FeCO axons

**B** Morphological classes postsynaptic to FeCO axons

**C** Overall distribution of morphological classes

**D** Neurotransmitter identity of neurons postsynaptic to FeCO axons

**E** Multisensory integration in postsynaptic neurons

**F** Schematic of FeCO connectivity

**G** Quantifying shared connectivity via cosine similarity

**H**

**I**

We found that the majority of synapses from claw and hook extension axons are onto local VNC interneurons and leg motor neurons (Fig. 2B, C). In contrast, more than half of all synapses from club axons are onto intersegmental neurons. Hook flexion axons are somewhere in between, making roughly similar proportions of their synapses onto local and intersegmental postsynaptic partners. Compared to the other FeCO subtypes, club axons also make a notably high number of synapses onto ascending neurons that convey leg somatosensory information to the brain.

We next used anatomical criteria to identify the developmental origins of all pre- and postsynaptic partners of FeCO axons (see

Methods). About 95% of adult neurons in the *Drosophila* VNC arise from 30 segmentally repeated neuroblasts (neural stem cells), each of which divides to form an 'A' and 'B' hemilineage[31]. Developmental hemilineages are an effective means to classify VNC cell types: neurons of the same hemilineage release the same primary neurotransmitter (i.e., Lacin's law)[32,33] and express similar transcription factors[34,35]. Previous research also suggests that neurons within a hemilineage are functionally related: thermogenetic activation of single hemilineages drove leg and wing movements[36], and connectome analyses of larval VNC neurons demonstrated that neurons within a hemilineage share common synaptic partners[37].

**Fig. 2 | FeCO neurons exhibit subtype-specific postsynaptic connectivity. A** We reconstructed all VNC neurons postsynaptic to FeCO axons from the front left leg (T1L) and divided them into morphological classes (examples provided). **B** Percent of synapses from each FeCO axon made onto VNC neurons of each morphological class. Top shows total number of output synapses made by each FeCO axon. **C** Per FeCO subtype, the total fraction of output synapses made onto each morphological class. **D** Proportion of total synapses made by each FeCO neuron onto neurons from different VNC hemilineages. **E** Heatmap shows the percent of neurons postsynaptic to a T1L FeCO subtype (as indicated along the rows) that also receive synaptic input from one of the following: T1L FeCO neurons, campaniform sensilla axons (CS), hair plate axons (HP), or bristle axons, and non-leg somatosensory neurons, including left neck chordotonal organ axons or left wing somatosensory axons. Neurons postsynaptic to claw and hook axons also integrate information from other leg proprioceptors including HP and CS axons. Neurons postsynaptic to club axons integrate information from wing and neck somatosensory axons. **F** Schematic of FeCO connectivity. **G** By querying the connectivity of each postsynaptic partner of each reconstructed FeCO neuron, we created (**H**) a connectivity matrix and (**I**) a cosine similarity matrix. **H** Connectivity matrix between FeCO axons and postsynaptic VNC neurons. FeCO axons are organized by morphological subtype and then by their cosine similarity scores. VNC neurons are organized by their cosine similarity scores. **I** Clustered pairwise cosine similarity matrices of all FeCO axons based on their postsynaptic connectivity. The cosine similarity between two neurons is the dot product of the normalized (unit) column weight vectors. If two FeCO neurons synapse with similar synaptic weights onto the same postsynaptic neuron, relative to the FeCO's total output, the pairwise cosine similarity is 1. Source data are provided as a Source Data file.

We found that club axons target neurons from different hemilineages than claw and hook axons (Fig. S3, see Supplementary Table 3 for links to view entire populations of each hemilineage in Neuroglancer). Neurons that are primarily postsynaptic to club axons come from hemilineages 0A/0B, 1B, 8B, 9A, 10B, and 23B. Of those, only 1B and 9A neurons receive any synaptic input from claw axons, but the connectivity is weak: ~6% and 0.5% of total FeCO input, respectively. Club and hook flexion axons target some shared hemilineages, including 1B, 9A, and 23B. Neurons from all other identified hemilineages are predominantly postsynaptic to claw or hook axons and do not receive any synaptic input from club axons. We further used the hemilineage designations to infer a neuron's primary neurotransmitter (Fig. 2D). The majority of *Drosophila* neurons release one of three primary neurotransmitters: acetylcholine, GABA, or glutamate[34,38]. In the fly, acetylcholine is typically excitatory, while GABA is typically inhibitory[34,39,40]. Glutamate is excitatory at the fly neuromuscular junction, acting on ionotropic glutamate receptors (GluRs), but is frequently inhibitory in the CNS, acting on the glutamate-gated chloride channel, GluCl[41]. Club axons synapse onto very few putative glutamatergic neurons compared to claw and hook axons (Fig. 2D).

We conducted similar analyses examining the presynaptic inputs to FeCO axons (Fig. S4). Generally, hook axons receive the most input synapses and have the most presynaptic partners, which include local, ascending, and intersegmental neurons (Fig. S4A, B). The majority of input synapses to FeCO axons are GABAergic (Fig. S4D). The strongest input comes from 9A neurons, which are primarily presynaptic to hook axons (Fig. S3C, D). Recent work found that a subset of 9A neurons suppress expected proprioceptive feedback from hook neurons during voluntary movements such as walking or grooming[29].

Together, these differences in postsynaptic connectivity suggest that claw and hook axons are connected to postsynaptic partners that are distinct from those downstream of club axons. These downstream partners differ in their morphology as well as their developmental stem-cell lineage. Hook and claw axon connectivity with local and motor neurons suggests that they play a role in fast feedback control of leg motor output. In contrast, club axons connect to intersegmental and ascending pathways that could relay leg vibration information to the brain to support detection of external mechanosensory signals (Fig. 2F). In support of this conclusion, we found that the VNC neurons that receive input from claw and hook axons also receive input from other leg proprioceptors, such as hair plate and campaniform sensilla neurons. In contrast, the VNC neurons that receive input from club axons receive little input from leg proprioceptors, and instead receive sizeable input from somatosensory neurons from the neck and wing (Fig. 2E, F).

## FeCO axons demonstrate subtype-specific downstream connectivity
We next investigated the specific postsynaptic partners targeted by claw, hook, and club axons, and the degree to which FeCO axons synapse onto distinct or overlapping circuits. First, we constructed a connectivity matrix to look at the postsynaptic connectivity of each FeCO neuron, organizing the rows of the matrix by FeCO subtype (Fig. 2G, H). Generally, postsynaptic connectivity is sparse, with each FeCO neuron contacting 21.1 ± 1.1 (mean ± s.e.m.) distinct postsynaptic partners (Fig. S1B). To quantify this connectivity structure, we calculated the cosine similarity score for pairs of FeCO axons based on their synaptic outputs (Fig. 2I; see "Methods"). Two FeCO axons have a high cosine similarity score if they make the same relative number of synapses onto the same postsynaptic neurons. Low similarity scores indicate either that two FeCO axons share few postsynaptic partners or that the relative number of synapses onto common postsynaptic partners is different.

Hierarchical clustering of cosine similarity scores confirmed that FeCO axons of the same subtype provide similar synaptic output to the same postsynaptic partners (Fig. 2I). FeCO axons tuned to different tibia positions (claw flexion vs. claw extension axons) or movement directions (hook flexion vs. hook extension axons) demonstrate very low (almost zero) cosine similarity scores, indicating that their postsynaptic connectivity is very different. Instead, hook and claw axons that share directional selectivity (claw and hook flexion or claw and hook extension axons) demonstrate some shared connectivity, as suggested by cosine similarity scores above zero. Unexpectedly, we found that hook flexion axons and club axons share some postsynaptic connectivity, as demonstrated by their relatively high cosine similarity scores and co-clustering. For example, one specific VNC interneuron received synaptic input from almost all club and hook flexion axons (Fig. 2H). We also calculated and clustered the similarity scores for FeCO axons based on their presynaptic inputs (Fig. S4E, F). However, because FeCO axons have far fewer (and in some cases zero) presynaptic partners (Fig. S1D, 2.9 ± 0.3 neurons, mean ± s.e.m.), these similarity scores are dominated by the shared connectivity of just a few presynaptic neurons. Claw extension and claw flexion axons receive little shared synaptic input. In contrast, hook flexion and hook extension axons all receive very similar synaptic input. Only a small number of club axons receive presynaptic input, but those that do exhibit high similarity to one another, except for two club axons whose upstream connectivity is more similar to that of hook axons.

In summary, FeCO axons demonstrate subtype-specific pre- and postsynaptic connectivity. FeCO axons within a subtype are generally more similar in their connectivity than FeCO axons of different subtypes, suggesting that information from each subtype is conveyed in parallel to different downstream neurons.

## Claw and hook axons connect directly and indirectly to leg motor neurons
So far, we have found that club axons synapse onto VNC neurons from different morphological classes and developmental hemilineages than the claw and hook axons. This segregated connectivity suggests that signals from club neurons are relayed to distinct downstream circuits with different functions than claw and hook neurons. Given that club neurons are the only subtype that respond to low-amplitude, high-

frequency vibration, we hypothesized that this distinct connectivity could reflect an exteroceptive function of club neurons compared to the proprioceptive function of claw and hook neurons. To explore this hypothesis, we next examined how each FeCO subtype connects to leg motor circuits.

FeCO axons can synapse directly onto motor neurons and can also indirectly excite or inhibit them via layers of intervening interneurons. The complexity of these feedback networks makes it challenging to infer how activity of FeCO neurons could impact leg motor neurons. To understand the general structure of these feedback networks, we first grouped leg motor neurons into motor modules[27] (Fig. 3B). Motor modules contain varying numbers of motor neurons that, based on their anatomy and presynaptic connectivity patterns, comprise a functional motor pool that drive a similar movement (e.g., tibia extension). We next examined the connectivity between FeCO neurons and motor modules in one of two ways. First, we plotted the overall number of synapses made between FeCO neurons, premotor interneurons, and motor neurons, inferring the interneurons' putative neurotransmitter according to their hemilineage assignment (Fig. S5). Second, we developed an impact score metric that summarizes this connectivity data in a single value by weighting direct and indirect connections between FeCO axons and motor modules, as well as the putative neurotransmitters of any intervening interneurons (Fig. 3A, C–E). This impact score is useful to understand trends or differences in motor connectivity among different subsets of FeCO neurons across multiple layers, but limited in its utility to predict neuronal activity as it does not include important factors such as circuit dynamics or intrinsic neural properties.

Both analyses revealed that claw and hook neurons make excitatory and inhibitory connections with many leg motor neurons. The pattern of their connectivity is consistent with previous recordings of motor neuron activity and optogenetic manipulations in *Drosophila*[25,28]. Claw and hook flexion axons provide excitatory feedback to motor neurons that extend the tibia and inhibitory feedback to motor neurons that flex the tibia. Claw and hook extension axons provide excitatory feedback to motor neurons that flex the tibia and strong inhibitory feedback to motor neurons that extend the tibia. Claw extension axons also provide excitatory feedback to other motor modules, such as the motor neurons that move the coxa forward (coxa promotor) and extend the trochanter. This connectivity suggests that FeCO feedback supports leg motor synergies that span multiple leg joints.

Consistent with our hypothesis that club neurons do not support local leg motor control, club axon connectivity with leg motor neurons is weak, demonstrated by a low impact score (Fig. 3D, note different scale bar). Club axons form no direct synapses onto leg motor neurons (Fig. S5E). However, they do indirectly and weakly connect to leg motor neurons innervating the long tendon muscle (LTM), which controls substrate grip[42] (Figs. S5E, 3D). Club axons also indirectly connect to the premotor peripherally synapsing interneuron (PSI) (Fig. 3E), which excites wing power muscles during takeoff[14,43,44]. This connectivity suggests a pathway by which activation of club neurons could lead to startle or escape behaviors, such as freezing and take-off.

Finally, we analyzed the overall structure of the connectivity of FeCO axons with premotor interneurons that synapse on leg MNs. We found that post-FeCO premotor interneurons primarily synapse onto a single motor module (Fig. 3F, G). This finding is consistent with previous work showing that most premotor neurons preferentially connect to specific motor modules[27]. We also found that all post-FeCO premotor neurons receive the majority of their synaptic input from only a single FeCO sensory subtype (Fig. 3F, G). This pattern of connectivity suggests that fly leg motor circuits have a modular organization, with dedicated interneurons connecting a single FeCO subtype with a single motor module (Fig. 3H).

## Club connectivity is consistent with a putative tonotopic map of tibia vibration frequency

Among the five FeCO subtypes, club axons stood out as separating into subclusters that had more shared connectivity with one another than other club axons (Fig. 2I). Past recordings of calcium activity from FeCO neurons in response to tibia vibration revealed that club axons are organized tonotopically[18,19].

We therefore wondered whether the connectivity subclusters we found in the VNC connectome could represent functional groupings of club axons tuned to similar vibration frequencies.

In support of this hypothesis, we found that the spatial organization of the connectivity subclusters of club axons reflects the tonotopy observed in prior experimental recordings. We discretized the connectivity subclusters into three groups (Fig. 4A) and found that they spatially tile the dorsal-ventral axis (Fig. 4B). Previous measurements of club axon activity in response to tibia vibration revealed a tonotopic map of frequency sensitivity along the anterolateral to posteromedial axis of the VNC, such that the most anterolateral axons respond most strongly to low frequency vibrations and the most posteromedial axons respond most strongly to high frequency vibrations[18]. Unfortunately, due to limitations in the orientation of the optical path, this previous data did not measure frequency tuning along the dorsal-ventral axis. However, comparing the anatomy of the axon subclusters we reconstructed in FANC to the images of calcium activity of club axons from Mamiya et al.[18] (Fig. 4C) suggests that the connectivity subclusters represent club axons with similar frequency tuning that also synapse onto common postsynaptic partners. By reconstructing an additional 14 club axons from the middle right leg (T2R), we found that this spatial organization is replicated in other leg neuromeres. Club axons from this leg also separate into subclusters that span the dorsal-ventral axis (Fig. 4A, B). Finally, we found that VNC neurons postsynaptic to club axons receive most of their input from club axons from the same connectivity subcluster across multiple legs (Fig. 4D, E). For example, club axons in the most dorsal subclusters of the T1L and T2R legs connect to overlapping downstream partners regardless of their leg of origin.

In summary, we found that club axons tile the dorsal-ventral axis and demonstrate overlapping postsynaptic connectivity with their immediate neighbors. We propose that individual club neurons in similar locations along the dorsal-ventral axis of each leg neuromere have similar vibration frequency tuning. If true, then our connectivity analyses suggest that postsynaptic neurons integrate information from club neurons with similar vibration tuning across different legs as well. Thus, the putative tonotopic structure observed in club axons would be preserved in postsynaptic neurons.

## Interneurons postsynaptic to club axons integrate information across legs

The major downstream partners of club neurons are interneurons from the 0A/0B, 8B, 9A, and 10B hemilineages (Fig. 5A). These interneurons express different primary neurotransmitters—8B and 10B are cholinergic, whereas 0A/0B and 9A are GABAergic. They also possess distinct morphologies that imply specialized roles in transforming club signals (Fig. 5B, C). Individual 10B interneurons primarily receive input from one leg and project to the contralateral and adjacent legs, whereas 8B interneurons arborize broadly and have mixed input and output synapses in all six neuromeres. 0A/0B interneurons project bilaterally and have pre- and postsynaptic sites on both the right and left sides of each VNC segment. 9A interneurons are the most localized, with their input and output synapses contained within a single neuromere. The diversity of these interneuron morphologies and connectivity suggests that club information is broadly relayed throughout the CNS through parallel pathways that integrate club information locally within a leg and globally across multiple legs. Integration of club signals within a leg could be important for amplification of weak vibration signals, while

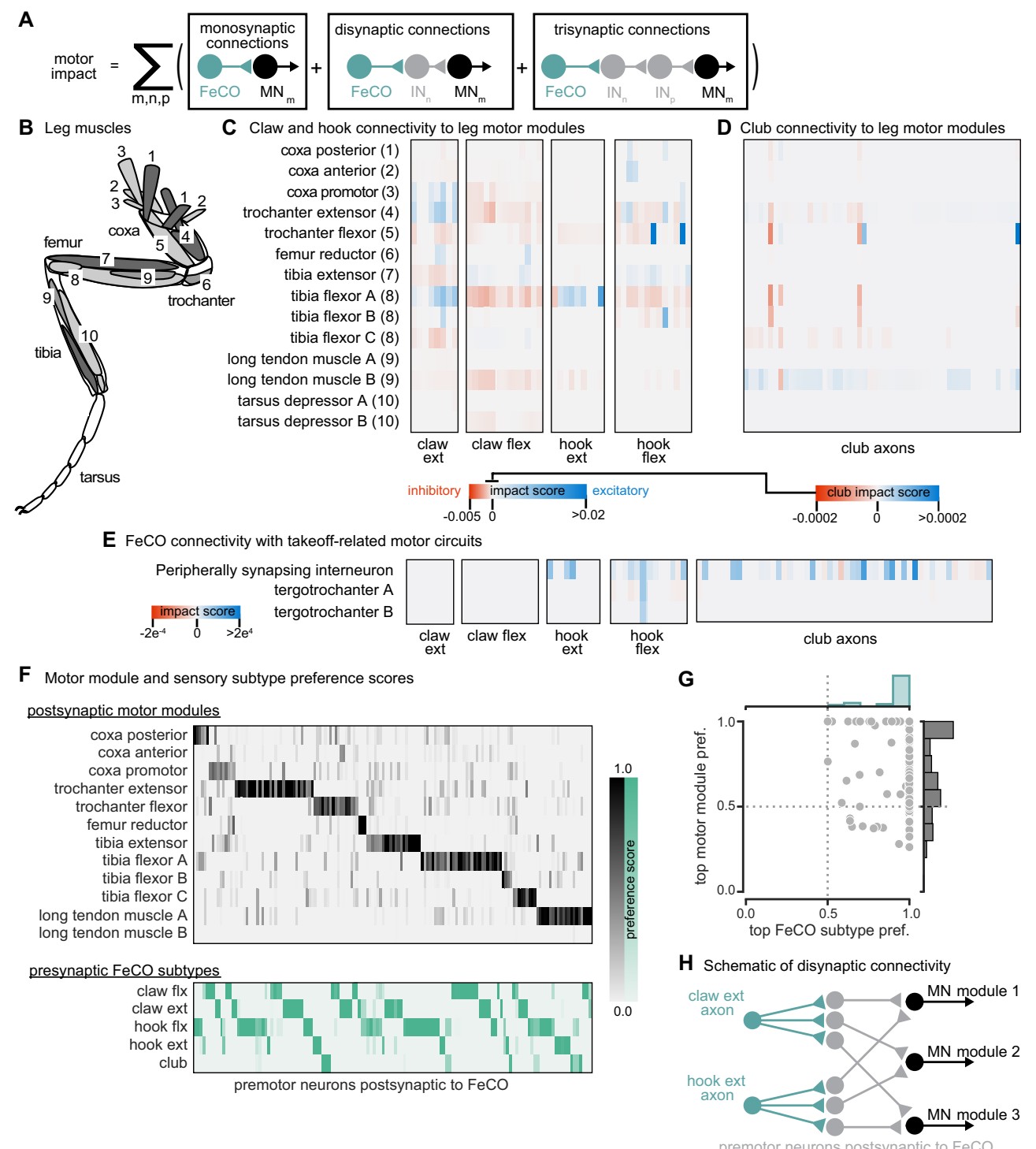

**Fig. 3 | The connectivity of hook and claw neurons is structured to impact activity of leg motor neurons. A** We developed an impact score that considers monosynaptic, disynaptic, and trisynaptic connections between FeCO axons and leg motor neurons (see "Methods"). **B** Schematic of the 18 muscles controlling the fly's front leg. Numbers correspond to motor module labels in (**C**). **C** Motor impact scores of claw and hook axons onto leg motor modules. Motor modules are functional groupings of motor neurons that receive common synaptic input and act on the same joint[2]. The target muscles of each motor module are indicated in (**B**). **D** Motor impact scores of club axons onto leg motor modules. Note the scale bar change from (**C**). **E** Motor impact scores of FeCO axons onto take-off related motor circuits. The peripherally synapsing interneuron is a premotor neuron involved in takeoff. The tergotrochanter is a leg muscle that is not active during walking but is

instead involved in jumping and takeoff[3–5]. **F** Motor module preference scores (gray, top) and FeCO subtype preference scores (green, bottom) for each premotor VNC neuron that receives input from FeCO axons (columns). Premotor neurons are arranged according to their preferred motor module followed by their preferred FeCO subtype. **G** Motor module preference (y-axis) plotted against FeCO subtype preference (x-axis) for each premotor VNC neuron that receives input from FeCO axons. **H** Schematic representation of the predominant connectivity pattern seen between FeCO neurons and motor modules. Premotor neurons postsynaptic to the FeCO are primarily dedicated to relaying information from a particular FeCO subtype to a particular motor module. Source data are provided as a Source Data file.

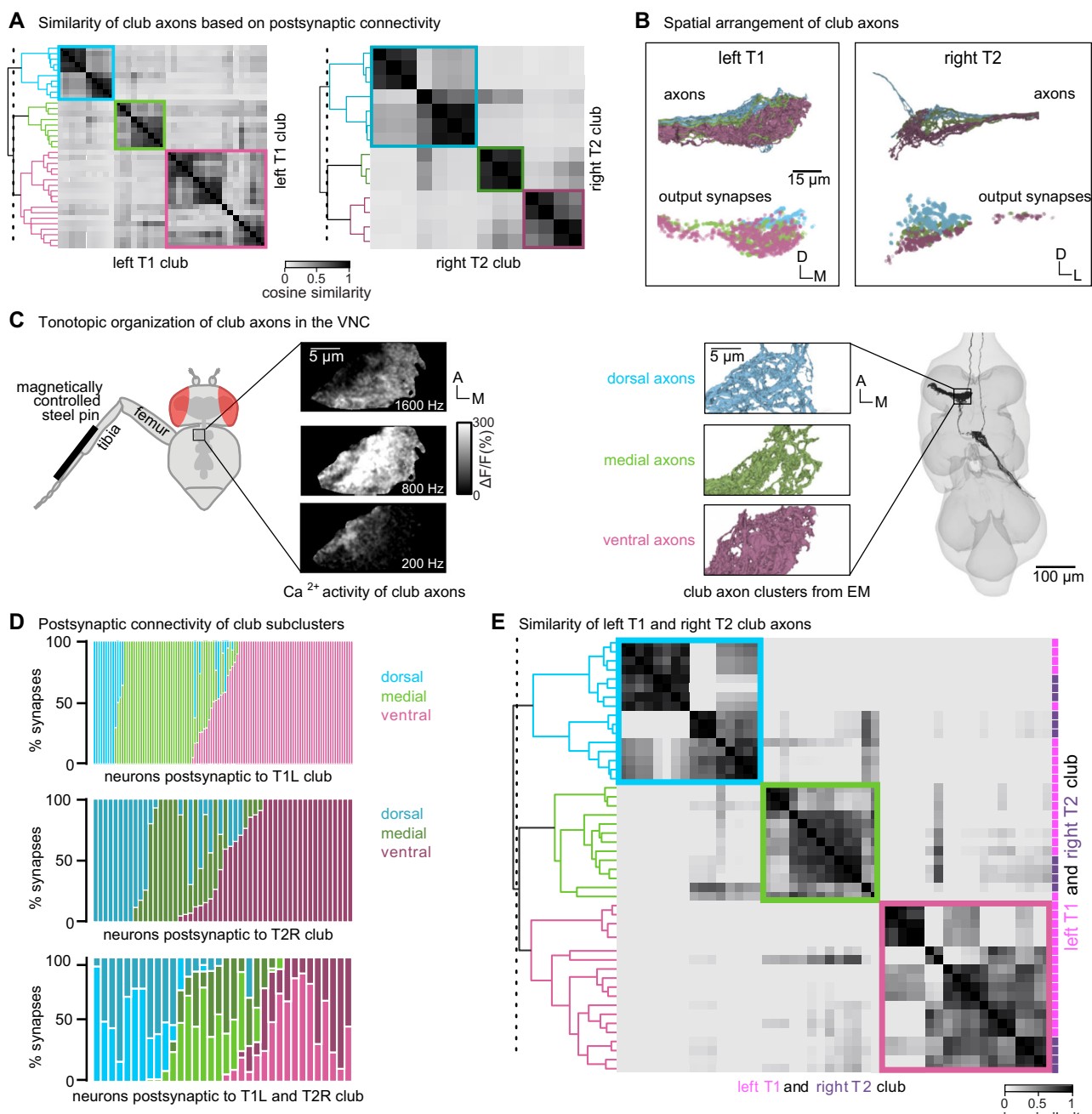

**Fig. 4 | Club neurons cluster into spatial groups that reflect the putative tonotopic map of tibia vibration frequency. A** Clustered pairwise cosine similarity matrices of the club axons based on postsynaptic connectivity. Sensory neurons with similar postsynaptic connectivity patterns cluster together, forming connectivity clusters. Left: Matrices for club axons in front left leg with connectivity clusters highlighted in blue (*n* = 10), green (*n* = 9), pink (*n* = 18). Right: Matrices for club axons in the middle right leg (T2R) with connectivity clusters highlighted in blue (*n* = 7), green (*n* = 3), pink (*n* = 4). **B** Top, Club axons within each connectivity cluster form 3 groups that span the dorsal-ventral axis: dorsal (blue), medial (green), and ventral (pink). Bottom, the spatial location of the output synapses for each of club neurons, color-coded by the corresponding connectivity cluster. **C** The dorsal-ventral organization of connectivity clusters is consistent with tonotopic mapping of tibia vibration frequency recorded from club axons with calcium imaging[6]. Left, Schematic of the experimental setup. Reprinted from Neuron,

vol. 111, Mamiya, A. et al., Biomechanical origins of proprioceptor feature selectivity and topographic maps in the *Drosophila* leg, 3230-3243.e14, Copyright (2023), with permission from Elsevier. Calcium data adapted from Mamiya et al.[18] depicting calcium responses from club axons to vibration frequencies (200 Hz, 800 Hz, 1600 Hz) applied to the tibia. In this experimental setup, the club axons are imaged from a single plane with the ventral side facing the objective. Right, reconstructed club axons in the FANC dataset separated by connectivity clusters, viewed from a similar plane as the calcium imaging data. **D** Fraction of input synapses from club neurons onto downstream partners. Club neurons are grouped based on connectivity cluster (dorsal: blue, medial: green, ventral: pink). **E** Clustered pairwise similarity matrices of the T1L (pink) and T2R (gray) club axons based on shared postsynaptic connectivity. Sensory neurons with similar postsynaptic connectivity patterns cluster together regardless of their leg of origin. Source data is provided as a Source Data file.

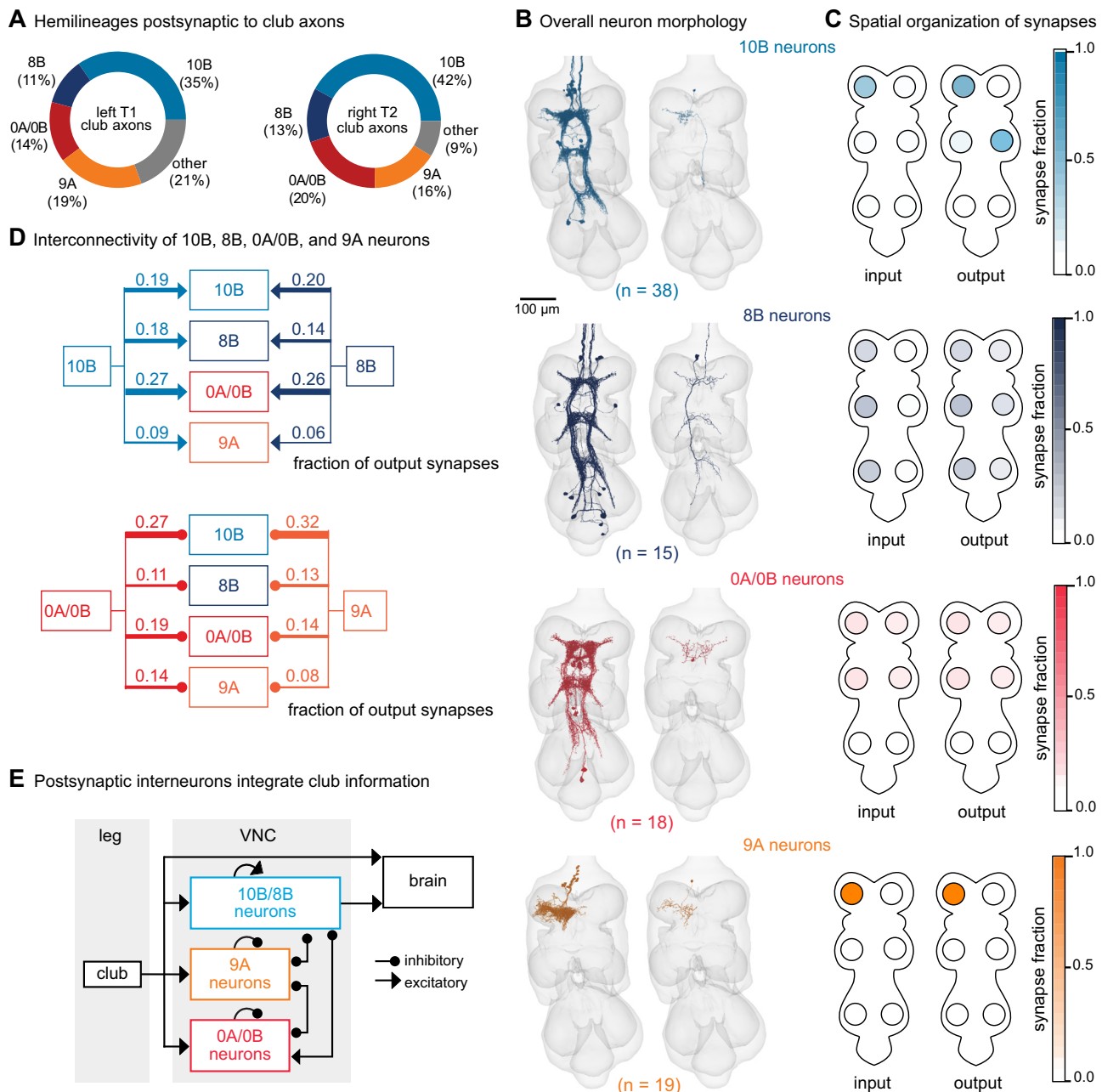

**Fig. 5 | VNC neurons integrate vibration signals from club neurons across multiple legs. A** Fraction of total output synapses from club neurons onto downstream interneurons separated by hemilineage class. 10B (light blue, *n* = 38), 8B (dark blue, *n* = 15), 0A/0B (red, *n* = 18), 9A (orange, *n* = 19). **B** Reconstructed interneurons downstream of left T1 club neurons from the FANC dataset. Left image shows all reconstructed neurons from each hemilineage that are downstream of T1L club neurons. Right image shows a single example neuron from that hemilineage. **C** Heatmap depicting the spatial locations of input (left) and output (right) synapses for 10B, 8B, 0A/0B, and 9A interneurons that are downstream of T1L club neurons. **D** Circuit diagram depicting recurrent connections between 10B, 8B, 0A/0B, and 9A interneurons. Each line indicates an excitatory or inhibitory connection and is labeled with the fraction of total output synapses each interneuron class makes with another interneuron class. **E** Schematic of the multi-layered connectivity downstream of club axons. Source data is provided as a Source Data file.

integration across legs could be important for spatial localization. Lateral and disinhibitory circuits may sculpt vibration information, for example, via normalization or gain control across the population.

We reconstructed all 0A/0B, 8B, 9A, and 10B interneurons that receive 4 or more synapses from the reconstructed T1L or T2R club neurons described above. Roughly half of the synaptic inputs onto 10B, 8B, 0A/0B, and 9A interneurons come directly from T1L/T2R club neurons or indirectly through their downstream (second-order or third-order) partners (Fig. S6). A much smaller fraction of inputs come from other leg or wing sensory neurons or other interneurons. The

remaining synaptic inputs come from neurons that have not yet been proofread. Since our proofreading efforts were focused on club neurons from the T1L and T2R legs, we predict that a significant proportion of the missing input comes from club neurons from other legs and their postsynaptic partners.

Finally, 10B, 8B, 0A/0B, and 9A interneurons downstream of club neurons exhibit high levels of recurrent connectivity among interneurons from different hemilineages and different legs (Fig. 5D). Thus, the circuitry downstream of club axons is recurrent and multi-layered (Fig. 5E). We speculate that this highly interconnected circuit

architecture supports the capacity to localize substrate vibrations in the external environment.

## Leg vibration information integrates with auditory and mechanosensory circuits in the brain

Vibration signals from club neurons are relayed to the brain by ascending club axons and ascending 8B and 10B interneurons (Fig. 6A). Since the FeCO has been implicated in sensing substrate vibrations for courtship and escape[23,24], we hypothesized that leg vibration information carried by ascending projections is integrated in the brain with sensory information from the antennae. The fly antenna also contains a chordotonal organ, known as the Johnston's Organ, which detects antennal displacements and local air vibrations[45–47]. To this end, we next identified where these ascending club, 8B, and 10B neurons project to in the brain and analyzed their downstream connectivity.

Within the FANC dataset, we proofread 8 ascending club axons from the T1L and T2R legs, 58 ascending 10B interneurons, and 52 ascending 8B interneurons (Fig. 6A, bottom). We then used the FlyWire brain connectome dataset[7,15] to identify ascending projections that matched light-level morphology of ascending projections from club axons, 8B interneurons, and 10B interneurons (Fig. 6A, top, and Supplementary Table 4 for links to view these neurons in Neuroglancer). Within the brain connectome, we found 24 axons that matched the projections of ascending club axons and 94 axons that matched the ascending projections of 8B and 10B interneurons. 8B/10B interneuron branching patterns in the brain are notably different from those of ascending club neurons. Club axons are smooth with few branches, while 8B/10B axons branch extensively. Due to the similarity of their ascending projections, we could not resolve which interneuron axons belong to which hemilineage (8B or 10B) in the brain connectome.

To understand the differences between these two ascending pathways, we compared their connectivity in the VNC and the brain (Fig. 6B). Ascending club axons and 8B/10B interneurons are interconnected and share several downstream partners in the VNC. For example, the majority of postsynaptic partners of ascending club axons in the VNC (17/19) and the brain (25/41) also receive input from ascending 8B/10B interneurons (Fig. 6B). However, 8B/10B interneurons have many more postsynaptic partners than club neurons, thus targeting many of the same postsynaptic partners as ascending club neurons, but also several non-overlapping downstream partners.

Consistent with our hypothesis that club neurons are exteroceptive, we found that ascending club and 8B/10B neurons target brain regions that broadly integrate external sensory information: the anterior ventrolateral protocerebrum (AVLP), wedge (WED), saddle (SAD), and gnathal ganglion (GNG) (Fig. 6C, D). The AVLP, WED, SAD, and GNG encode mechanosensory information from the antennae, including signals related to wind and courtship song[45,48–50]. Neurons in the WED encode antennal vibration, are tonotopically organized, and some even respond to high frequency antennal vibrations (>600 Hz)[45]. Ascending club neurons and 8B/10B interneurons primarily converge onto downstream partners that also receive input from other head mechanosensory neurons (Fig. 6E), including auditory and mechanosensory Johnston's Organ neuron (JON) subtypes, mechanosensory bristles, and antennal campaniform sensilla (Fig. S7). This shared connectivity suggests that, in the brain, neurons integrate mechanosensory information from across the body, including the legs, wings, neck, head, and antennae. We speculate that this comparison of sensory information across the body could contribute to the detection and localization of mechanical vibrations in the external environment (Fig. 6F).

## Discussion

Here, we used connectomic reconstruction of neural circuits to infer the function of limb somatosensory neurons from patterns of synaptic connectivity. Prior experiments in *Drosophila* and

other insects had suggested that the femoral chordotonal organ (FeCO) serves a dual proprioceptive and exteroceptive function[18–24]. Our analyses support this conclusion and suggest that each function is supported by distinct FeCO sensory subtypes connected to distinct downstream circuits. Based on their connectivity, movement-sensing claw and hook neurons are primarily proprioceptive, providing feedback to local leg motor circuits. In contrast, vibration-sensing club neurons are primarily exteroceptive. Club axons provide feedback to intersegmental circuits that integrate somatosensory information across multiple limbs and then convey that information to the brain. These analyses demonstrate the power of connectomic mapping and analysis to identify putative functions of somatosensory neurons. They also motivate future work to test the function of circuits for limb proprioception and exteroception in behaving flies.

## Role of the FeCO in local leg motor control

We found that movement- and position-sensing hook and claw axons synapse directly and indirectly onto leg motor neurons (Fig. 3). This connectivity is consistent with prior evidence that the FeCO contributes to stabilization of leg posture[20,25,28]. We also found that claw and hook axons provide feedback to motor neurons controlling movement about multiple joints, consistent with work in locusts[51] and wētās[52].

Proprioceptive feedback needs to be flexibly tuned to support shifting behavioral demands[53]. For example, during voluntary movement, proprioceptive pathways promoting stabilizing reflexes may be attenuated to avoid opposing the intended movement. One possible mechanism underlying this context-dependent tuning is presynaptic inhibition of sensory axons[54,55]. In support of this mechanism, we found several inhibitory upstream partners of claw and hook axons (Figure S3). We also showed in a recent study that hook (but not claw) axons are presynaptically inhibited during voluntary leg movement[29]. In addition to direct feedback onto somatosensory axons, proprioceptive feedback may also be tuned via context-dependent inhibition of downstream pathways.

Finally, we found that claw and hook axons synapse onto a small number of intersegmental and ascending neurons (Fig. 2). Intersegmental projections could relay proprioceptive information to the motor circuits of other legs. However, past work suggests that feedback from the FeCO of one leg does not strongly affect control of other legs—manipulating activity of FeCO neurons has little effect on inter-leg coordination[56–59]. Ascending neurons that are postsynaptic to claw and hook neurons could relay leg proprioceptive information to the brain to inform action selection. Calcium imaging experiments have shown that many ascending neurons are active during behaviors like walking[60]. Additionally, neurons in higher order visual areas and in the central complex encode walking stride, speed, and turning behavior even in the absence of visual input, suggesting that they receive self-motion cues from the legs[61–63].

## Tonotopic and spatial organization of club axons

Individual club neurons are tuned to specific vibration frequencies, collectively forming a tonotopic map in the VNC[18]. We found that club axons are spatially organized into sub-clusters with shared postsynaptic connectivity that tile the dorsal-ventral axis of the VNC (Fig. 4). By comparing this spatial organization to prior recordings of club axon activity in the VNC[18], we hypothesize that the dorsal club axons respond to higher frequencies while the ventral club axons respond to lower frequencies. Actual measurements of frequency sensitivity along the dorsal-ventral axis would be necessary to confirm this hypothesis. Intersegmental second-order neurons receive input from club neurons originating in different legs but situated in a similar location along this dorsal-ventral axis, suggesting that this putative tonotopy is conserved in downstream circuits. However, many of these

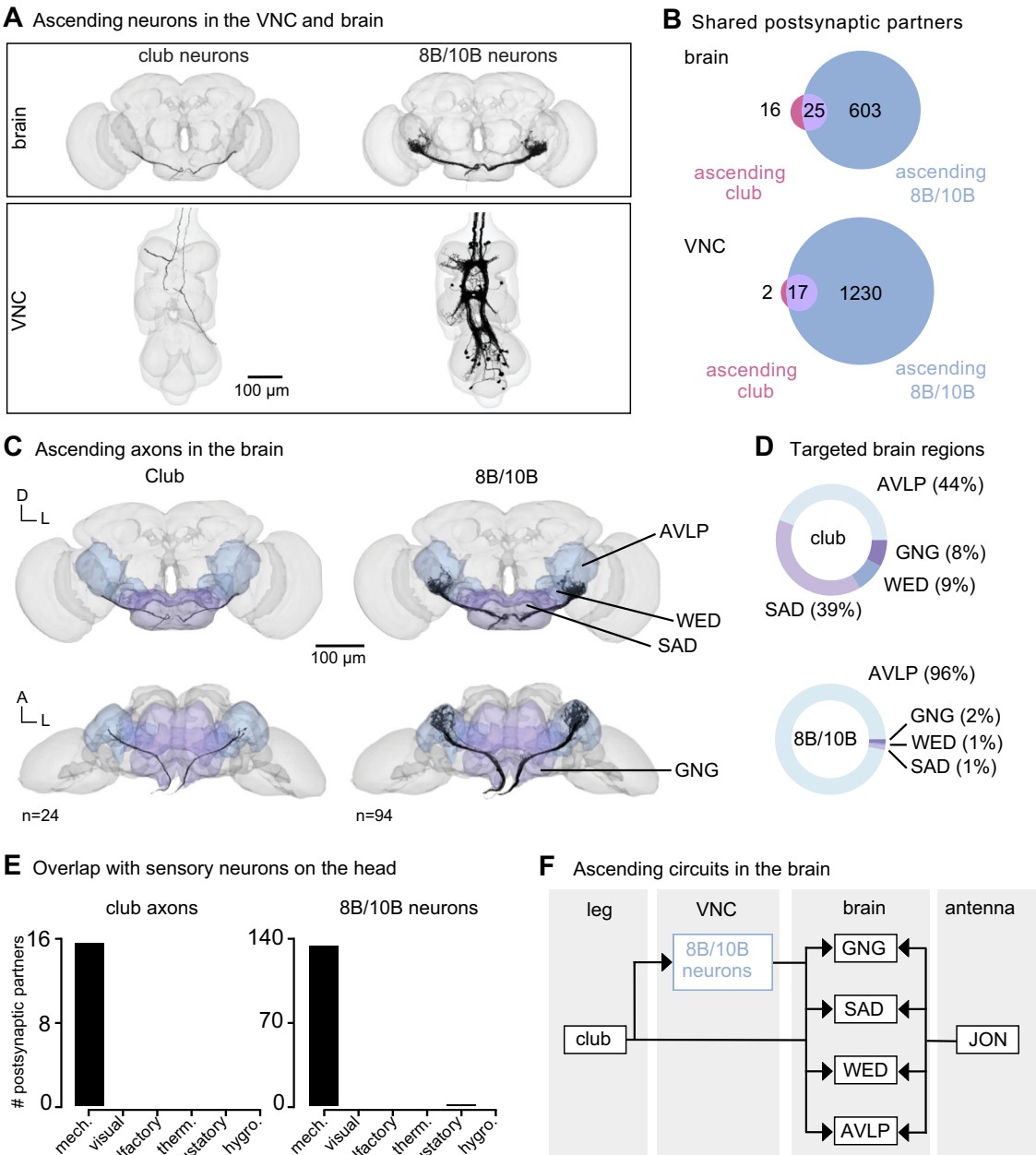

**Fig. 6 | Vibration signals from club neurons are transmitted directly and indirectly to the brain and integrated with mechanosensory signals from the antenna. A** Ascending club axons and ascending 8B and 10B interneurons that are reconstructed in the FANC (bottom, 8 club axons and 52 interneurons) and FlyWire (top, 24 sensory axons and 94 interneuron axons) datasets. **B** Venn diagrams of shared postsynaptic partners between ascending club neurons and ascending 8B/10B neurons in the VNC (top) and brain (bottom). **C** Images of the ascending club axons (*n* = 24) and ascending 8B and 10B interneurons (*n* = 94) in the brain dataset with targeted brain regions highlighted (FlyWire). Axons project to the anterior ventrolateral protocerebrum (AVLP), wedge (WED), saddle (SAD), and gnathal ganglion (GNG). **D** Percentage of synaptic outputs from ascending club axons (top) and ascending 8B/10B interneurons (bottom) in each brain region. The ascending club axons and ascending interneurons differ with respect to distribution of output synapse location. **E** Number of postsynaptic partners of ascending club axons (left) and ascending 8Bs/10Bs (right) that are shared with other sensory neurons in the brain. **F** Circuit diagram depicting the projection patterns of ascending club and ascending 8B/10B interneurons in the brain, which integrate with antennal mechanosensory circuits. Source data is provided as a Source Data file.

second-order neurons are densely interconnected. We hypothesize that this multi-layered circuit could support spatial localization of vibration stimuli. Based on this hypothesis, if we were to record the activity of 10B or 8B neurons while applying vibration stimuli to different locations around the fly, we would expect to find individual neurons tuned to specific frequencies and spatial locations. Our reconstruction efforts here were limited to club neurons in only two leg neuropils and their postsynaptic partners. The similarity of connectivity across the T1L and T2R club circuits suggests that our findings will likely hold true for the club axons from the other legs.

## Putative exteroceptive function of club neurons

Our analyses support the hypothesis that club neurons primarily function as vibration-sensing exteroceptors. Interestingly, the sensitivity of club neurons is similar to Pacinian corpuscle low threshold mechanoreceptors (LTMRs) in mammals (40–1000 Hz). Like club neurons, Pacinian LTMRs are active during a wide variety of natural behaviors, including walking and grooming, as well as during substrate vibration[3]. Recent work in the mouse has found that Pacinian signals converge with auditory input from the cochlea in the lateral cortex of the inferior colliculus (LCIC)[64]. LCIC neurons respond more strongly to

coincident vibration-auditory stimulation than to either stimulus alone. The Pacinian to LCIC circuitry in the mouse resembles the club to AMMC circuitry that we describe here in the fly, suggesting that integration of limb vibration and auditory signals may be a common principal of mechanosensory processing across diverse animals.

How animals use these vibration signals in natural environments remains unclear. In *Drosophila*, males produce both airborne and substrate-borne vibrations as part of courtship[65–69]. Genetic silencing of FeCO neurons in female flies reduces their receptivity to male courtship song[23], suggesting that the FeCO is involved in courtship. Consistent with this hypothesis, we found downstream neurons in the brain that integrate vibration information from both legs and antennae (Fig. 6). Vibration-sensitive club neurons in the leg and Johnston's Organ neurons in the antennae respond to overlapping vibration frequencies[18,19,45–47], but the full sensitivity range for each group has not been carefully measured. We hypothesize that the integration of vibration information from the antennae and legs could guide courtship behavior, for example by comparing the quality of airborne and substrate-borne courtship communication signals. Importantly, in this study, we reconstructed the FeCO circuits within a female VNC and brain. If club neurons support detection of courtship-related signals, the circuitry downstream of the club neurons could be sexually dimorphic and thus differ in male flies. Further reconstruction in the male VNC connectome (MANC) would be needed to test this possibility.

Aside from courtship, vibration information from club neurons could also be used to detect movements of predators or other threats, such as wind or rain. Consistent with this hypothesis, we found that club axons are indirectly connected to escape-related neurons, including motor neurons innervating the long tendon muscle (LTM) and the premotor peripherally synapsing interneuron (PSI) (Fig. 3). Activation of club neurons could promote leg freezing via activation of the LTM, or take-off via activation of the PSI. Previous studies have implicated the FeCO in escape and courtship responses in multiple insect species beyond *Drosophila*[21,23,24]. Crustaceans also have leg chordotonal organs that can detect external substrate vibrations which have been implicated in supporting vibration-based social communications[70–72]. Many insect species also possess subgenual organs, specialized vibration sensors in the tibia, but flies lack these sensory structures[73]. Thus, in *Drosophila*, club neurons are likely the primary sensors that detect substrate vibrations via the legs.

### Lack of convergence across FeCO subtypes in second-order neurons

We were surprised to find that the downstream connectivity of each FeCO subtype is quite distinct: very few VNC neurons receive synapses from more than one FeCO subtype (Fig. 2). Past work proposed a higher degree of convergence across FeCO subtypes within second-order neurons. Using whole cell patch-clamp recordings and 2-photon calcium imaging, we previously characterized multiple VNC interneuron cell types that encode combinations of femur-tibia joint movement, position, and vibration, suggesting that they receive input from multiple FeCO subtypes[25]. In another study, we combined optogenetic activation and calcium imaging to map the functional connectivity between FeCO axons and their downstream partners[26]. That study also found examples of VNC interneurons that receive inputs from more than one FeCO subtype. One possible explanation for this discrepancy is that our connectome analyses predominantly focused on direct connections between FeCO neurons and their synaptic partners. The integration of information from multiple FeCO subtypes could be via indirect connections involving multiple intervening interneurons. In addition, Agrawal et al.[25] found evidence for mixed electrical and chemical synapses that connect FeCO sensory neurons with some downstream partners. The FANC EM dataset was not imaged at sufficient spatial resolution to resolve electrical synapses.

Finally, we did find some weak shared connectivity between FeCO cell types that share directional selectivity, such as claw and hook flexion axons or claw and hook extension axons. Due to the adventitious nature of their physiology experiments, Agrawal et al.[25] and Chen et al.[26] may have, by chance, characterized the few interneurons that do indeed receive synaptic information from multiple FeCO subtypes. We did find some overlap in the connectivity of hook flexion and club axons (Fig. 2). This finding is consistent with Agrawal et al.[25], who found one cell type, 9Aa neurons, that respond to both flexion and vibration of the femur-tibia joint. However, the implications of this overlap remain to be investigated. For example, integrating exteroceptive and proprioceptive signals could enable the fly to determine if the source of the vibration is due to movement of its leg. Alternatively, flies could concurrently sense movement and vibration information from the leg to assess substrate texture.

In addition, we found substantial overlap in the downstream connectivity of FeCO neurons and other leg proprioceptive neurons, such as campaniform sensilla (CS) and hair plates (HP) (Fig. 2E). In fact, claw and hook axons shared a larger number of downstream partners with CS or HP neurons than with other FeCO subtypes. Work from stick insects and other species suggests that such multimodal input is important for context-dependent control of proprioceptive reflexes[74,75]. For example, signals from load-sensing CS neurons can reduce the effect of FeCO activation on leg motor neurons[74].

### Looking forward

Connectome analysis is a powerful tool to generate and falsify hypotheses about circuit function. Thanks to advances in serial-section electron microscopy and automated image segmentation, we are close to having multiple connectomes of the fruit fly brain and VNC. As more neurons within these connectomes are connected to specific functions, such as motor neurons that control a particular joint or sensory neurons that detect specific signals, these maps become increasingly useful anatomical frameworks for generating hypotheses about the neural control of behavior. Though physiological and behavioral measurements are still necessary in order to determine how a circuit functions, our study illustrates how a global view of synaptic connectivity can reveal organizing principles that motivate future experiments.

## Methods

### Reconstruction of neurons in the FANC connectome

Neurons in the Female Adult Nerve Cord (FANC) electron microscopy dataset[8] were previously segmented in an automated manner[14]. To manually correct the automated segmentation of our neurons of interest, we used Google's collaborative Neuroglancer interface[76]. Many of the FeCO axons in T1L were previously identified[8], and most of the claw and hook axons were previously partially corrected[29]. Here, we identified and corrected additional claw axons as well as all club axons in T1L and T2R. Identification was guided by light-level images of FeCO subtype-specific genetic driver lines[18,19]. An FeCO neuron was deemed "completed" in its reconstruction if all major branches were attached as confirmed by comparison to light microscopy images. All uncertain connections were double-checked by at least two experienced proofreaders, and then a final check of each neuron was completed at the end to again confirm that no false connections were added to a neuron and no major branches were missing.

To reconstruct pre- and postsynaptic partners of FeCO neurons, we identified all objects in the automated segmentation that received at least 4 synapses from an FeCO neuron or made at least 3 synapses onto an FeCO neuron. Synapses were detected automatically as described by Azevedo et al.[14]. Past work[15,30] found that applying a 3–4 synapse threshold mitigates the inclusion of false positive connections. We then proofread those objects until associated with either a cell body or an identified descending or sensory process. A small

number of objects were categorized as fragment segments and could not be connected to a cell body or an identified descending or sensory process. We deemed a neuron as "proofread" once its cell body was attached, all major branches reconstructed, along with as many additional branches as could be confidently attached[77]. Neuron annotations were managed by CAVE, the Connectome Annotation Versioning Engine[78]. We used custom Python scripts to interact with CAVE via CAVEclient[78].

We additionally reconstructed a subset of third-order neurons that are 2 hops away from T1L and/or T2R club neurons. First, we identified all objects in the automated segmentation that receive at least 4 synapses from an 8B or 10B neuron that is postsynaptic to a club neuron. We chose to focus on neurons postsynaptic to intersegmental 8B and 10B neurons to gain greater proofreading coverage of club circuitry in other legs.

### Novel partners analysis

To identify the number of new postsynaptic partners added to our dataset per each FeCO sensory neuron we reconstructed, we first found all postsynaptic partners of all reconstructed T1L FeCO axons. Then, we randomly sampled the FeCO neurons one at a time (without replacement) in a cumulative fashion, and calculated how many novel postsynaptic partners were connected to each additional FeCO neuron. We re-did this random sampling 50 times. To extrapolate the resulting curve and estimate the likely number of postsynaptic partners for the entire 152 neurons in the FeCO, we used the curve_fit() function from the SciPy python package to fit a logarithmic function.

### Cosine similarity scores

Cosine similarity (for example, Fig. 2I) was calculated using the cosine similarity method from the scikit-learn python package. Cosine similarity scores were then hierarchically clustered using the agglomerative clustering methods from the scikit-learn python package.

### Branch preference scores

Using K-means, we clustered all output synapses from a given FeCO subtype based on their Euclidean distance from one another. We formed 3 clusters, each of which corresponds to a major branch. We then determined a branch preference score for every postsynaptic neuron by dividing the number of synapses the postsynaptic neuron received from one branch by the total number of synapses it received from all branches. In Fig. S2A-D, we plot these preference scores on ternary plots. Each postsynaptic neuron is represented by a point whose size varies according to the total number of synapses that that neuron receives from that FeCO subtype. In Fig. S2E, we randomly subsampled the synapses (while maintaining the clusters as identified above). We then repeated the above analysis, but only plotted the strongest preference scores per each postsynaptic neuron.

### Definition of cell classes

Neurons pre- and postsynaptic to FeCO axons were identified as motor, sensory, ascending, descending, intersegmental, or local neurons. Motor neurons have a cell body in the VNC and a process in the leg nerve. These neurons were recently identified in the FANC dataset for the front left leg[14,27]. Sensory neurons have a process in the leg nerve but no cell body in the VNC. Ascending neurons have a process in the neck connective and a cell body in the VNC. Descending neurons have a process in the neck connective but no cell body in the VNC. Intersegmental and local neurons have a cell body and all processes in the VNC. The processes of intersegmental neurons spanned multiple neuromeres, whereas those of local neurons were contained in a single neuromere. All pre- and postsynaptic neurons were manually checked to make sure they were in the correct categories.

### Identification of hemilineages

In *Drosophila*, neurons that share a developmental origin (i.e., belong to the same hemilineage) possess common anatomical features[36] and release the same fast-acting neurotransmitter (e.g., GABA, glutamate, or acetylcholine)[32]. We took advantage of this knowledge to identify the hemilineage of each neuron upstream and downstream of FeCO axons in the FANC connectome. We first identified and grouped together local, intersegmental, and ascending VNC neurons based on where their primary neurite entered into the neuropil. These groups of similar primary neurites were then identified as known hemilineages using light microscopy images of sparse GAL4 lines, cell body position along the dorsal-ventral axis[32,36,79,80], and personal communication (James W. Truman, David Shepherd, Haluk Lacin, and Elizabeth Marin). Putative neurotransmitter was then assigned by referencing Lacin et al.[32]. Not all of the clues are available for all of the neurite bundles. See Supplementary Table 3 for links to view entire populations of each hemilineage in Neuroglancer, an online tool for viewing connectomics datasets[76].

### Motor impact score

A presynaptic neuron's monosynaptic impact score onto a postsynaptic neuron is defined as the number of synapses made by the presynaptic neuron onto the postsynaptic neuron, divided by the total number of input synapses received by the postsynaptic neuron. Then, based on the presynaptic neurons' putative neurotransmitter according to its hemilineage assignment, this impact score is either considered excitatory (positive) or inhibitory (negative). In the fly, acetylcholine is typically excitatory, while GABA is typically inhibitory[34,39,40]. Glutamate is excitatory at the fly neuromuscular junction, acting on ionotropic glutamate receptors (GluRs), but is frequently inhibitory in the CNS, acting on the glutamate-gated chloride channel, GluCl[41].

To compute the motor impact score of a given FeCO neuron onto a motor module (Fig. 3), we summed together the calculated impact scores of monosynaptic connections, disynaptic connections, and trisynaptic connections between the FeCO neuron and all motor neurons (MNs) within a module. The impact score of monosynaptic connections between an FeCO neuron and a motor module is as described above, but summed across all MNs within a module. We assume that direct FeCO input to MNs would be cholinergic, and thus excitatory.

For the impact score of a disynaptic connection, we multiplied the monosynaptic impact score from an FeCO neuron onto neurons that are postsynaptic to the FeCO and presynaptic to MNs from the specific motor module (postFeCO/preMN neurons) by the impact score of those postFeCO/preMN neurons onto the MNs within a module. If the postFeCO/preMN neuron was identified as cholinergic, then this disynaptic impact score was considered to be excitatory/positive, and if it was identified as GABAergic or glutamatergic, then it was considered to be inhibitory/negative. We then summed together all disynaptic impact scores from the FeCO neuron to the MNs of a module.

For the impact score of a trisynaptic connection, we first found all neurons with an identified hemilineage that were postsynaptic to the FeCO neuron (postFeCOs) and all neurons with an identified hemilineage that were presynaptic to the MNs within the relevant module (preMNs). We then multiplied the monosynaptic impact score from the FeCO neuron onto a postFeCO neuron by the impact score of the postFeCO neuron onto a preMN neuron, and this was multiplied by the impact score of the preMN neuron onto the MNs within a module. If both the postFeCO and preMN neurons were excitatory or both inhibitory, then this trisynaptic impact score was positive. If one neuron was inhibitory and one was excitatory, then this trisynaptic impact score was negative. We then summed together all trisynaptic impact scores from the FeCO neuron to the MNs of a module.

## Module preference score

To compute the preference score for a motor module (Fig. 3), we summed the number of synapses onto each MN within a module (as defined by Lesser et al.[27]) and divided by the total synapses onto all MNs. To compute the sensory subtype preference score for a FeCO subtype (Fig. 3), we summed the number of synapses received from all FeCO neurons of a given subtype and divided by the total synapses received from all FeCO neurons.

## Circuit analysis in the FAFB/FlyWire connectome

To study connectivity in the brain, we used the Full Adult Fly Brain connectome (FAFB[11]) reconstructed and proofread by the FlyWire community[11,15,78,81]. All data are from public release version 783.

## Identification of ascending neurons in the FAFB/FlyWire connectome

First, we manually screened through the repository of Gen1 MCFO images on FlyLight[80] for candidate images of VNCs that exhibit hallmark expression of the ascending club axons, ascending 8B interneurons, and ascending 10B interneurons in the VNC. To identify the anatomy of the ascending projections in the brain, we matched the ascending axons and interneurons in the VNC to the corresponding images in the brain. Next, we matched the anatomy of the ascending projections in the brain based on the light-level images to the FAFB dataset using flywire.ai[81] and the Codex platform[82]. Specifically, we queried neurons classified as ascending and cholinergic[83–85], then matched candidates to the light-level images of the target neurons. See Supplementary Table 4 for links to view the ascending neurons in Neuroglancer[76].

## Reporting summary

Further information on research design is available in the Nature Portfolio Reporting Summary linked to this article.

# Data availability

Data presented in the paper was analyzed from the CAVE materialization v604 timestamp 1684915801.222989 and the Connections Princeton No Threshold synaptic connectivity table from the public FlyWire Connectome Dataset v783: https://codex.flywire.ai/api/download. Annotated connectivity matrices (Fig. 2) are available as Python Pandas data frames (https://pandas.pydata.org/) at the GitHub repository: https://github.com/sagrawal/Lee_2024. Links to public segmentations are available throughout the text, as well as in a document at the GitHub repository. Source data are provided with this paper.

# Code availability

Scripts to recreate the analyses and figures in the paper, as well as scripts to recreate the connectivity matrices for users authorized to interact with the CAVEclient are available at Github: https://github.com/sagrawal/Lee_2024. All analysis was performed in Python 3.9 using custom code, making extensive use of CAVEclient: https://github.com/seung-lab/CAVEclient and CloudVolume to interact with data infrastructure, and libraries Matplotlib, Numpy, Pandas, Scikit-learn, Scipy, stats-models, and VTK for general computation, machine learning, and data visualization. Additional code is available at https://github.com/htem/FANC_auto_recon, providing additional tutorials, code, and documentation for interacting with FANC and joining the FANC community.

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

## Acknowledgements

We thank members of the Tuthill laboratory for technical assistance and feedback on the manuscript. We also thank Jasper S. Phelps, Wei-Chung Allen Lee, and the FANC community for their contributions to the proofreading of the VNC connectome, as well as Forrest Collman and other members of the CAVE management system at the Allen Institute for Brain Science. We thank Leila Elabbady, Ellen Lesser, Shirin Mohammadian, Gwendolyn Swannell, and Brandon Pratt for permission to use their unpublished reconstructions of sensory neurons in FANC. We thank Jim Truman, David Shepherd, Haluk Lacin, and Elizabeth Marin for assistance with hemilineage identification. This work was supported by a Postdoctoral Research Fellowship from the Deutsche Forschungsgemeinschaft (DFG, German Research Foundation) project 432196121 to C.J.D, a Searle Scholar Award, a Klingenstein-Simons Fellowship, a Pew Biomedical Scholar Award, a McKnight Scholar Award, a Sloan Research Fellowship, the New York Stem Cell Foundation, and NIH grants R01NS102333, R01NS128785, and U19NS104655 to J.C.T., NIH grants K99NS117657 and R00NS117657 to S.A, and NIH grant T32 NS 99578-3 to S.-Y.J.L. and J.C.T. J.C.T. is a New York Stem Cell Foundation—Robertson Investigator.

## Author contributions

S.-Y.J.L., C.J.D., J.C.T., and S.A. conceived the project. J.C.T. and S.A. acquired funding. S-Y.J.L., C.J.D., A.C., and S.A. proofread neurons in FANC. S.-Y.J.L., C.J.D., S.A. analyzed data. S.-Y.J.L., C.J.D., J.C.T., and S.A. wrote the paper with input from A.C.

## Competing interests

The authors declare no competing interests.
