## [Transparent Peer Review file · Nature Communications]

Divergent neural circuits for proprioceptive and exteroceptive sensing of the *Drosophila* leg

Corresponding Author: Dr Sweta Agrawal

Version 0:

Reviewer comments:

Reviewer #1

(Remarks to the Author)

The manuscript titled "Divergent neural circuits for proprioceptive and exteroceptive sensing of the *Drosophila* leg" by Lee et al. has been greatly improved after revisions. The authors have sufficiently addressed the concerns raised by this reviewer. I have just a few minor concerns that I outline below. Once these are addressed I recommend acceptance of the article.

1. The urls in the Supplemental Tables did not work for me. The authors should ensure the links are accessible prior to publication.
2. In the Figure Legend of Fig 2 and the diagram of Fig 2F, what is the rationale for concluding that club neurons primarily contact intersegmental and ascending neurons? The % outputs onto local neurons (18.0%) is comparable to that onto ascending neurons (19.5%).
3. In Fig. 6D, is there a "club" label missing inside the top circle?
4. Fig. 4C is still somewhat confusing relative to the text. The figure layout makes it appear that the top calcium image corresponds to the activity of dorsal neurons and bottom to ventral neurons (which the text clarifies is not the case). Adding an anatomical direction label (ie, anterior/lateral) to the calcium images would help.

(Remarks on code availability)

The code appears appropriate. It contains a README file and comments throughout. It will be useful to others in recreating the paper's figures as well as analyzing similar datasets.

Reviewer #2

(Remarks to the Author)

My high-level concerns have largely been addressed with the authors' edits and reply to reviewers. The 'motor impact' metric is now properly caveated (lines 233-4) and the addition of Sankey plots is a useful addition. The methodological procedures used during reconstruction are also now better described, as are caveats due to imperfect preservation of sensory neurons. The addition of Figure S2 provides evidence of robustness of results to non-saturating (i.e. backbone-only) reconstructions. Overall the paper is substantially improved and provides solid results that will be of interest to the connectomics and motor circuit research communities.

My remaining concern arises from a portion of the authors' reply to reviewers. The authors write, "We attached all major branches of the FeCO neurons presented in this study (confirmed by comparison to light microscopy images) as part of our reconstruction efforts. These major branches are generally referred to as the 'backbone' of the neuron. However, there are also many fine-scale branches that are not as easily resolved by light microscopy, which are referred to as 'twigs.'"

However, the now-cited Schneider-Mizell et al. paper (line 531, reference 79) does not define backbones as major branches easily resolvable by light microscopy; rather, they are branches (sometimes quite fine) that contain microtubules. I am unaware of any direct comparison between microtubule-containing backbones and light-microscopy resolved branches; in particular I am not certain they are 1:1 equivalent. I am left wondering whether the authors really reconstructed all neurites containing microtubules, or whether they simply reconstructed the 'big stuff', compared to light microscopy images of the neurons, and post-hoc called these backbones.

Could the authors clarify this point? This question is unlikely to affect the main findings of the paper but precision in reported reconstruction methods will be important to any downstream users of the authors' data, and consistency in the use of the backbone/twig terminology should also be maintained in the literature.

(Remarks on code availability)

Reviewer #3

(Remarks to the Author)

(Remarks on code availability)

We thank the reviewers for their comments on the revised manuscript. We agree with essentially all of their remaining concerns and have made efforts to edit our manuscript accordingly. Please see below for a point by point response to each comment.

Reviewer #1:

1. The urls in the Supplemental Tables did not work for me. The authors should ensure the links are accessible prior to publication.

We apologize – it looks like the links were broken when we converted the document to a pdf. We will double check that the links are functional in the final proofs.

2. In the Figure Legend of Fig 2 and the diagram of Fig 2F, what is the rationale for concluding that club neurons primarily contact intersegmental and ascending neurons? The % outputs onto local neurons (18.0%) is comparable to that onto ascending neurons (19.5%).

You are correct, the club neurons do not synapse onto very many more ascending neurons than local neurons. However, club neurons synapse onto many more ascending neurons than any of the other FeCO subtypes (19.5% compared to 1.4-8.2%). We have revised the text to clarify.

3. In Fig. 6D, is there a “club” label missing inside the top circle?

Yes - this is now fixed in the updated version.

4. Fig. 4C is still somewhat confusing relative to the text. The figure layout makes it appear that the top calcium image corresponds to the activity of dorsal neurons and bottom to ventral neurons (which the text clarifies is not the case). Adding an anatomical direction label (ie, anterior/lateral) to the calcium images would help.

In the discussion, we hypothesize that the dorsal neurons respond to higher frequencies while the ventral neurons respond to lower frequencies, which is in line with the figure layout.

Unfortunately, due to limitations in the orientation of the optical path, the previously obtained calcium imaging data could not measure frequency-tuning along the dorsal-ventral axis, but rather did so along the anterolateral to posteromedial axis. As a result, the calcium imaging data cannot be perfectly aligned with the dorsal-ventral clusters from the EM data. As suggested, we added the axes labels to Fig 4C to help orient the reader.

Reviewer #2:

My remaining concern arises from a portion of the authors' reply to reviewers. The authors write, "We attached all major branches of the FeCO neurons presented in this study (confirmed by

comparison to light microscopy images) as part of our reconstruction efforts. These major branches are generally referred to as the 'backbone' of the neuron. However, there are also many fine-scale branches that are not as easily resolved by light microscopy, which are referred to as 'twigs.'"

However, the now-cited Schneider-Mizell et al. paper (line 531, reference 79) does not define backbones as major branches easily resolvable by light microscopy; rather, they are branches (sometimes quite fine) that contain microtubules. I am unaware of any direct comparison between microtubule-containing backbones and light-microscopy resolved branches; in particular I am not certain they are 1:1 equivalent. I am left wondering whether the authors really reconstructed all neurites containing microtubules, or whether they simply reconstructed the 'big stuff', compared to light microscopy images of the neurons, and post-hoc called these backbones. Could the authors clarify this point? This question is unlikely to affect the main findings of the paper but precision in reported reconstruction methods will be important to any downstream users of the authors' data, and consistency in the use of the backbone/twig terminology should also be maintained in the literature.

We agree with the reviewer that consistency in the use of the backbone/twig terminology should be maintained in the literature. As a result, we have modified the manuscript to remove all mention of the term "backbone" as we did not use presence of microtubules as a guiding criteria in our reconstructions. We instead focused on major branches based either on light microscopy images, similarity to other reconstructed VNC neurons, or our best judgements.